# The Efficacy of Primavera, a Prevention Programme on Alcohol and Tobacco Use among 10–12-Year-Old Schoolchildren: A Randomized Controlled Cluster Study

**DOI:** 10.3390/ijerph18083852

**Published:** 2021-04-07

**Authors:** Cristina Diaz Gomez, Alain Morel, Isabelle Sedano, Henri-Jean Aubin

**Affiliations:** 1French Monitoring Centre for Drugs and Drug Addiction, 75018 Paris, France; cristina.diaz-gomez@ofdt.fr; 2Oppelia NGO, 75012 Paris, France; amorel@oppelia.fr; 3CSAPA Horizon 02, 02100 Saint Quentin, France; isedano@oppelia.fr; 4Centre de Recherche en Epidémiologie et Santé des Populations (CESP), French Institute of Health and Medical Research (INSERM) U 1018, University Paris-Saclay, Hopital Paul Brousse AP-HP, 94800 Villejuif, France

**Keywords:** alcohol, tobacco, prevention, adolescents, psychosocial skills, randomized controlled cluster trial

## Abstract

Alcohol and tobacco use is a major health problem and one of the first causes of the burden of disease and mortality. School-based alcohol and tobacco use prevention programmes that have demonstrated efficacy are most often based on psychosocial skill development, individuals’ experiential learning strategies, and community resources. Furthermore, early and prolonged interventions have been recommended. Primavera is a pluri-annual, generic, multimodal, experiential-oriented prevention program. It runs over a three-year period from the last year of primary school to the second year of secondary school. This randomized controlled cluster study aimed at assessing the effects of the Primavera programme compared to a control prevention intervention among schoolchildren from 10 to 12 years in eight secondary schools in a particular French geographical area. The primary outcomes were lifetime tobacco use and past-month alcohol use. Data were collected at baseline and over three follow-up time points. In all, 287 and 266 questionnaires, respectively, were collected at baseline from the Primavera group and from the control group. Attrition was 45% and 41%, respectively. The SARS-COV2 pandemic crisis made it impossible for questionnaires to be collected during the final year. After adjustment, children from the Primavera group were less likely to report current alcohol use at the end of the first year (odds ratio = 0.39, 95% CI: 0.18–0.78) and past-month alcohol use at the end of the second year (odds ratio = 0.07, 95% CI: 0.01–0.66) compared to those from the control group. The results for psychosocial skills and alcohol and tobacco use denormalization were contrasted. Primavera is shown to be effective in reducing alcohol use among schoolchildren.

## 1. Introduction

Alcohol and tobacco use are among the greatest risk factors for the global burden of disease and for premature mortality [1,2,3,4]. Prevalence data show that alcohol and tobacco use start at an early age, whilst precocity has been associated with an increased risk to develop substance use disorders [5]. According to the EnCLASS survey carried out in France in 2018 among 20,000 middle and high school students aged 11 to 18, alcohol is the first psychoactive substance children experiment with [6]. The number of students found to have drunk at least one alcoholic beverage in their life in Grade 6 (44.3%) clearly indicates that this initiation to alcohol begins in primary school. However, less than one in ten middle school students in Grade 6 (8.6%) reported that they had been drinking in the past month. The number of students using tobacco is significantly lower at the beginning of the middle school (1.4% last month smoking), although 7.6% reported that they had already smoked. As with lifetime use of alcohol, the “middle school years” seem to mainly be a phase where young adolescents are introduced to tobacco. For the two substances studied, boys experimented more than girls. Alcohol is still the first substance used during adolescence, followed by tobacco. Moving from Grade 9 (last grade of middle school) to Grade 10 (first grade of high school) is characterized by pursuing experimenting. As is true for middle school, alcohol is the most commonly used substance among high school students (85.0%), considerably more than tobacco (53.0%).

Substance use motives among young people are manifold. Alcohol and tobacco use behaviours are related not only to individual factors, such as poor awareness of drug-use-related risks, but also to the social context young people find themselves in; this is particularly important in shaping or in failing to shape their self-esteem and their ability to develop favourable health behaviours. For prevention actions to be efficacious in improving risks awareness and social skills, it is important to implement early interventions; it has been suggested that preventive actions among young people should be initiated early on in their lives [7,8,9,10]. Continuity in action over several school years, especially during the transition between primary and secondary school, could also be a factor predictive of efficacy [8,9,11,12,13,14,15]. Concerning the specific qualities that make prevention programmes at school efficacious among schoolchildren, a few studies have shown the benefits of a theoretical base integrating the promotion of health and the development of psychosocial skills through the use of individuals’ experiences and resources derived from their environment [8,10,11,16,17,18,19,20,21]. In addition, the training of relevant staff and the integration of the educational community, parents in particular, could have a favourable impact on prevention interventions [8,11,17,19].

In response to this, the Primavera programme was developed with the following features: school-based, pluri-annual, universal, generic and multimodal, and based on experiential learning strategies. The main objective of this randomized controlled study was to assess the efficacy of the Primavera school-based programme for preventing youth alcohol and tobacco use.

## 2. Materials and Methods

### 2.1. Type of Study

The design of the present study was a randomized controlled cluster study [22]. The study compared two groups of schools. The intervention group received the Primavera programme, which was to be deployed over a three-year period in seven two-hour sessions. The first three sessions took place during the last year of primary school and the next four sessions were carried out in secondary school (three sessions in the first year and the last session in the second year). The control group received a non-specific health promotion and prevention intervention in a two hour session, for students in their last year of primary school. The name given to this control intervention was “Cap Santé”, which could translate as “Heading for health”. Assessment measures were the same at baseline and across the three follow-up time points respectively at 6 months, 18 months, and 30 months after the start of the Primavera and Cap Santé interventions. The baseline assessment took place in the last year of primary school, before the start of the intervention; the first follow-up time point also occurred in the last year of primary school, but after the intervention. The second assessment endpoint took place during the first year of secondary school, before the summer holidays. The last follow-up time point was due to take place 30 months after the start of the intervention. However, the SARS-COV2 pandemic crisis made it impossible to carry out the last assessment, at 30 months from the launch of the intervention (spring 2020).

### 2.2. Randomization

The randomization unit was conducted during secondary school. Randomization of the school classes or schoolchildren was not used in order to limit a potential bias of interindividual influence. Indeed, as the Primavera prevention programme aimed to have a direct impact on the schoolchildren’s educational environment (class, school, and home), random allocation between intervention and control groups applied to schoolchildren from the same classroom or the same school would have created biased assessment conditions (the intervention would have had an impact on the control group). Clustered randomization is often used in this type of assessment for similar reasons [19,23,24,25,26]. To have a sufficiently large pool of participants, a sample of twelve secondary state schools from the Amiens academy was constituted over a restricted geographical zone. The sample was composed of schools in a geographical area within a 30 km radius of the prevention programme centre (Saint-Quentin, France). Among them, the four schools most distant from the restricted geographical zone were excluded, to optimize commuting for the prevention programme organizers. In all, eight secondary schools were randomized, providing a theoretical pool of 826 schoolchildren in first-year secondary school, corresponding to a recruiting pool of 41 primary schools. Randomization was carried out using the randomized block method. Two blocks of four secondary schools were computer generated. Allocation was 1:1. Figure 1 shows a flow chart of the recruitment process for the schools and the participants.

### 2.3. Study Population

The study population was composed of children aged 10 to 12 attending school from the last year of primary school to the second year of secondary school. All school classes in the randomized schools were included in the research.

### 2.4. Intervention

Primavera is a multimodal, generic, universal, pluri-annual, school-based prevention programme, based on an experiential approach. It runs over three years from the final year of primary school to the second year of secondary school and, therefore, requires continuity in the intervention between primary and secondary schools in the same geographical zone.

The programme results from the merger of several prevention programmes: a programme of Quebec origin “GymSAT” for schoolchildren aged 8–10 years and a programme “A vos marques, prêts, Santé!” for schoolchildren aged 10–11 years; these two programmes aimed to develop psychosocial skills. Another programme called “ICCAR” for secondary school children (12–13 years), focused on risky and addictive behaviour, completed the final framework. The merger of these three programmes gave birth to Primavera.

The Primavera prevention strategy uses health promotion as a reference basis [27] and is mainly based on experiential learning [17,28] via the development of psychosocial skills, deployed over seven sessions with a facilitator trained in the implementation of the programme. For this study, the first three sessions in primary school and the final session in secondary school were led by the designers of the Primavera programme. The team was composed of a nurse and a health promotion provider. The four intermediate sessions were led by the school educational team (teachers or school nurses), who had been trained to use the programme. A detailed description is found in the Supplementary Material.

The Primavera programme is based on a pedagogical and theoretical framework to be applied to all sites, while conditions and tools, can be adapted locally by the pedagogical teams. The pedagogical and theoretical framework is based on experiential learning and health promotion. It aims at developing a participative pedagogical method, in order to integrate the children’s environment into the programme as a whole. The programme should devote a maximum amount of time to workshops per class and per year. The programme aims to reinforce psychosocial skills, in particular self-esteem, self-efficacy, motivation, adaptation strategies, and empowerment.

The experiential approach, thus, encourages subjects to think about their way of life, about what they consume and what makes—or prevents them—to consume, the negative and positive consequences of these behaviours, their ability to act to find individual and collective satisfaction and well-being, taking into account the physiological and psychosocial limitations that characterize each individual. From their own experience and experience shared within a group, individuals become aware of their sources of satisfaction, enabling them to find pleasure in life, avoid pitfalls, and remain free to make their own choices. The experiential approach, thus, aims to provide subjects with tools that will help them, particularly in terms of prevention, to understand their own life experience, in their environment, and to become their own life experts. The workshops have been developed precisely to provide an educational experience dimension to guide each individual.

To enable an evaluation of the objectives specifically followed by an experiential approach, training the Primavera programme organizers involves key notions such as the importance of a common language in the educational community, the definition of substances, the potential harmfulness of substances and the satisfaction they provide, the different modes of use, and risk and protection factors.

Workshops with children are centred on three themes:the body, pleasure and suffering, emotions and their management;screens/social networks/games/tobacco/alcohol; andwhat influences choice: advantages, risks, limits, others.

Tools in the shape of games have been developed to conduct the workshops. They are presented to the educational teams during the training sessions; the different schools can choose the tools that will be the most suitable.

The Primavera programme includes 23 h of activities per year, i.e., 3 h of meetings aiming to involve the educational community, 12 h of training (the basis of the experiential approach, health promotion, organizational methodology, and implementation) for the adults (teachers and school nurses) wishing to become workshop facilitators, six one-hour workshops with the students in homogeneous classes to deploy the programme, and finally 2 h of evaluation of the programme.

### 2.5. Ethical Aspects

The participants were all minors. Informed consent signed by the parents or the legal guardians was collected. To ensure schoolchildren’s protection, an agreement was signed by the Amiens academy and the research team, stipulating the practical conditions for recruiting in agreement with the relevant national education authorities. An information leaflet was addressed to the schoolchildren’s parents, to inform them about the research and to ask for their signed informed consent for their children to take part in the study. The study was notified to the French national commission for the protection of personal data and privacy (CNIL no. 1954657 v 0 from 26 April 2016).

### 2.6. Primary Outcomes

The Primavera programme aims to avoid or to delay (1) lifetime prevalence of substance use and (2) transition from the first experiment towards frequent use of any psychoactive substance. However, only alcohol and tobacco were targeted in the assessment carried out for the present research. It seems that these two substances are by far those that, at the beginning of secondary school, signal the onset of potentially addictive behaviour [29,30,31].

The research was based on the hypothesis of a reduction of tobacco and/or alcohol use reported by schoolchildren in their second year of secondary school in the intervention group, compared to the control group. Two primary outcomes were thus retained: (1) the consumption of an alcoholic beverage (champagne, cider, wine, beer, strong alcohol, and pre-mixed drinks) in the last 30 days before the survey was carried out (last month alcohol use) and (2) the lifetime prevalence of smoking at least one cigarette, cigar, or pipe (those who have ever smoked). The questions, taken from the World Health Organization (WHO) Health Behaviour in School-Aged Children (HBSC) questionnaire [32,33], were the following:In the last 30 days, have you drunk any alcohol?In your lifetime, have you ever smoked tobacco (at least one cigarette, one cigar, or a pipe)?

### 2.7. Secondary Outcomes

Essentially based on indicators from the WHO HBSC survey [32,33], the secondary outcomes were as follows: the current (more often than once in lifetime but less frequent than once last 30 days) use of alcohol, whatever the type of alcoholic beverage; the use of tobacco within the past month, and denormalization of tobacco or alcohol use, at school and at home; peer support (PS) and the development of psychosocial skills (PSS) in the children.

Current alcohol use was assessed with the following question: “how often do you usually drink alcoholic beverages?” The question was addressed to the children who had reported having drunk one or more of the following alcoholic beverages: beer, wine, strong alcohol, pre-mixed drinks, cider, champagne, or other drinks containing alcohol. A composite indicator was calculated grouping the children’s responses, whatever the alcoholic drink, according to whether they said their consumption of alcohol was “every day”, “every week”, “every month”, or “rarely” (not “never”).

The use of tobacco in the past month was estimated on the basis of the children’s responses on cigarette smoking. The question was the following: “In the last 30 days, how many cigarettes have you smoked (if you have ever smoked)?”

The denormalization of tobacco/or alcohol use in the children’s environment (i.e., aiming to make the use of the substance less acceptable) was assessed through the following question: “If you ever use or used tobacco/or alcohol, would the following people (“your best friend”, “other children of your age at school”, “your father”, “your mother”) be happy with that?” Five responses were proposed: “completely agree”, “rather agree”, “neither agree nor disagree”, “rather disagree”, “disagree completely”. The variable of interest grouped the children’s responses expressing disagreement (“rather disagree” or “completely disagree”). For each substance, two composite indicators were measured. The first indicator aimed to explore the denormalization of tobacco/or alcohol in the children’s school environment. It gathered responses concerning the best friend and other schoolmates. The second indicator concerned the family home. It gathered responses about the attitudes of the children’s mother and father.

The indicators relating to peer support (PS) were measured by asking the children to say to what extent they agreed with the three following items concerning their school class:PS1: The students in my class like being together.PS2: Most students in my class are nice and willing to help others.PS3: The other students accept me as I am.

The questionnaire used a Likert scale identical to the one used to assess the denormalization variable; we grouped the responses of those who were in agreement (“completely agree” or “rather agree”).

The psychosocial skills (PSS) were assessed with six statements:PSS1: When I do not agree with a friend, I tell him/her.PSS2: When I have problems, I ask my friends for help.PSS3: Once I have decided to do something, I see it through.PSS4: When I make decisions, I think about the pros and cons.PSS5: I try to solve my problems without starting a fight.PSS6: When a friend gives me a piece of information, I check whether it is true.

For each of the above statements, the variable of interest grouped the following responses: “most often”, “often”, or “sometimes”. The Likert scale used also included two other possible responses: “almost never” and “never”.

### 2.8. Calculation of the Sample Size

It was hypothesized that among children in their second year of secondary school consuming alcohol in the past month before the survey there would be a difference of 28% in the intervention group compared to the control group, basing this hypothesis on a systematic review [34]. Furthermore, the calculation of the numbers was estimated on the assumption that the proportion of drinkers in the past month in their second year of secondary school would be 14% among subjects assigned to the control group, whereas the percentage would be 10% in the intervention group, i.e., an expected gain of 28%.

The sample size calculated was, thus, 314 subjects per group for an α risk of 5% corrected for multiple tests (Bonferroni’s correction) by 2 (number of primary outcomes) [35] and for a type II error of 10% (power of 90%). Anticipating a percentage of loss to follow-up of 20%, the objective was to recruit 393 subjects per group, i.e., 786 subjects for the two groups.

### 2.9. Data Collection

Data collection was carried out using a self-administered, anonymous, paper questionnaire. The questionnaires did not carry any personal identification data, as the subjects were not individually followed. The data collected relating to personal life included schoolchildren and family characteristics and sociodemographic situation (gender, month, and year of the students’ birth and family situation), and the use of alcoholic drinks and tobacco (lifetime use, quantities consumed, place of their first substance consumption, parental alcohol and tobacco use, psychosocial skills, and perception of individual health status). The questionnaire displayed the French national commission for the protection of personal data and privacy’s (CNIL) approval, informing the students about the right of the parents or other representatives of parental authority to access, rectify, or delete data. An independent polling company administered the questionnaire, and an external data capture company collected the data.

### 2.10. Statistical Analyses

The analysis strategy comprised three phases, one for each assessment: (1) a bivariate logistic regression model, comparing in turn all the variables (variables of the primary and secondary outcomes, predefined adjustment variables, potentially confounding variables) between the two intervention groups; (2) multivariate logistic regression model A1, comparing primary and secondary outcomes between the two intervention groups, adjusting on predefined adjustment variables and on other variables with an odds ratio (OR) significantly different from 1 in the bivariate logistic regression for the follow-up time point considered; (3) multivariate logistic regression model A2, identical to model A1 but with the introduction of all the other primary and secondary outcomes. However, the “use of tobacco in the past month” variable in the A2 multivariate model had to be removed, because the number of students who responded positively to this item in all assessment time points was too small (the conditions for the validity of multivariate logistic analyses require at least 5–10 events per explanatory variable in the model).

The predefined adjustment variables were the following: gender, age, three social-economic variables (individual bedroom, two cars or more in the household, trip abroad during the last holiday), and a weighting variable for each school to take the magnitude of the numbers analysed for each school into account. The other potential confounding variables are presented in Appendix A.

Comparisons were carried out separately at inclusion (baseline), at 6 months from the start of the interventions (follow-up time point 1), and at 18 months (follow-up time point 2). As follow-up time point 3 could not be completed because of the SARS-COV2 pandemic, the comparison that was initially planned at the end of the prevention programme at 30 months from its launch could not take place.

An α risk threshold of 5% was retained. The risk linked to the multiplicity of tests was taken into consideration by using multivariate regression models, enabling two increasing levels of adjustment to be included. As the volume of missing data was under 5% in the questionnaires collected, there was no imputation procedure.

## 3. Results

An initial sample of twelve secondary schools was constituted. Four of them were excluded as they did not meet the inclusion criteria because of their remoteness from the priority geographical zone. The eight remaining secondary schools were randomized and divided into two groups of four secondary schools (control and intervention), corresponding, respectively, to 19 and 22 primary schools in the area providing the intake of the secondary schools. The study enabled the collection of 266 questionnaires in the control group and 287 questionnaires in the intervention group completed by students in the final year of the primary schools retained (Figure 1). The numbers included were below the sample size of 314 subjects calculated per group (85% of the questionnaires collected in the control group versus 91% in the Primavera group). Ten questionnaires were discarded for incoherence in the data management phase. Attrition was low at 6 months of follow-up (3.8% attrition in the control group and 3.5% attrition in the Primavera group). The attrition rate, however, reached over 50% of the initial sample at 18 months follow-up (48% in the control group and 54% in the Primavera group).

A description of the responses to the questionnaires in the Primavera and control groups is shown in Table 1 (primary and secondary outcomes) and in Appendix A (for all other variables, see Appendix A). As a brief overview, at baseline, the students were aged 10 to 11 on average, boys accounted for 52% of the sample. Ever smoking was reported by 6% (Primavera) and 6.8% (control) of the students, past-month alcohol use was reported by 7.1% (Primavera) and 5.4% (control) of the students, past-month tobacco use prevalence was 1.8% (Primavera) and 2.7% (control), and the current alcohol use was reported by 25.3% (Primavera) and 21.3% (control) of the students.

Bivariate intergroup comparisons showed significant differences at each assessment time point. At baseline, psychosocial skills (PSS2, 4, 5, and 6), outings to parks and museums, and no more than 2 h spent watching screens during the week, were significantly less frequently reported, whereas good school results and parental alcohol consumption were more frequently reported in the Primavera group than in the control group. At follow-up time point 1, outings to parks and museums and under two hours watching screens in the week were significantly less often reported in the Primavera group than in the control group; however, good school results; talking about alcohol at home with parents; and remembering having received information in school about alcohol, tobacco, or another substance were significantly more often reported in the Primavera group than in the control group. At follow-up time point 2, psychosocial skills (PSS3) were significantly less often reported in the Primavera group than in the control group, while having a parent using tobacco or alcohol and remembering having received information in school about alcohol, tobacco, or another substance were significantly more often reported in the Primavera group than in the control group. It can be noted that in Appendix A the intergroup difference concerning the recall of prevention actions continued to increase from the baseline to the last assessment time point.

For the primary outcome of alcohol use in the last 30 days, adjustment models A1 (ORa1) and A2 (ORa2) showed that the children from the Primavera group were less likely to report past-month alcohol use compared to those from the control group at 18 months after the start of the intervention (ORa1 = 0.10 [95% CI: 0.02–0.42]; ORa2 = 0.07 [95% CI: 0.01–0.66]). For the secondary outcome of current alcohol use, adjustment models A1 and A2 showed that the children from the Primavera group were less likely to report a current use of alcohol than those from the control group at 6 months after the start of the intervention (ORa1 = 0.50 [95% CI: 0.27–0.88]; ORa2 = 0.39 [95% CI: 0.18–0.78]). The imbalance in favour of negative responses for tobacco use in the last 30 days was such that multivariate models A1 and A2 could not be applied to the three assessment time points. The A1 adjustment model did not alter the intergroup differences found in the bivariate regression models concerning psychosocial skills. However, after applying adjustment model A2, only PSS2 remained significantly less likely to be reported in the Primavera group than in the control group at baseline. Despite its nonsignificance after adjustment A2, the intergroup difference in PSS4–6 psychosocial skills at baseline was no longer observed at subsequent follow-up time points. Although nonsignificant after adjustment A2, PSS3 was less likely to be reported in the Primavera group than in the control group at follow-up time point 2. Concerning the outcome of denormalization of substance use in the children’s school environment, tobacco consumption was more strongly disapproved, and alcohol consumption was less disapproved in the Primavera group than in the control group, after A2 adjustment at follow-up time point 1. The imbalance in favour of positive responses concerning the denormalization of tobacco and alcohol at home was such that multivariate models A1 and A2 could not be applied. Finally, concerning peer support, no difference was found between the two groups, whatever the adjustment model and the assessment wave.

## 4. Discussion

This randomized controlled cluster study enabled the efficacy of the Primavera programme to be demonstrated among schoolchildren at the end of primary and the beginning of secondary school. The efficacy of the programme was demonstrated on one of the primary outcomes, past-month alcohol use, after adjustment during follow-up time point 2 (ORa2 = 0.07 [95% CI: 0.01–0.66]), at the end of the first year of secondary school, i.e., at the end of the second year of the prevention programme. The probability of experimenting with tobacco (second primary endpoint) was not different between the group exposed to the Primavera programme and that exposed to the control programme. The effect of the Primavera programme compared to the control programme was more contrasted for the secondary outcomes: lesser probability of current alcohol use at follow-up time point 1 (end of primary school) but poorer PSS3 psychosocial skills for “When I have decided to do something, I see it through” at follow-up time point 2, a difference that was not found after complete A2 adjustment. However, it is important to note that the students from the Primavera group had poorer baseline psychosocial skills PSS2 and PSS4–6 (including the complete adjustment model for PSS2) before the start of the Primavera or the control interventions, differences that disappeared after the start of the prevention interventions. An improvement in psychosocial skills PSS2 could, thus, reasonably be attributed to the Primavera programme (“When I have problems, I ask my friends for help”). Despite the fact that an adjusted comparison could not be conducted for the denormalization of alcohol and tobacco use at home, intergroup differences concerning the denormalization at school were found at follow-up time point 1, with greater denormalization of tobacco use but a lesser denormalization for alcohol use in the Primavera group compared to the control group.

This is, to our knowledge, the first published randomized controlled study on a universal school-based prevention programme on alcohol and tobacco use and psychosocial skill development in France. The Primavera programme met most of the recommended criteria for the development of prevention programmes against alcohol and tobacco use in schools: beyond the need to address substance use issues, it aims more widely to develop psychosocial skills [7,8,11,17,19,21], it is multimodal [8,11,17,19] and pluri-annual [9,12]. In addition, the programme is implemented across the primary–secondary school transition, a critical period in terms of emotional, social, and academic adjustment [13,14,15]. This programme is in current use in French schools by several trained teams. Qualitative feedback from participant schools shows that Primavera is well received and accepted by educational teams, children, and parents. For example, teachers reported that, following their experience with the Primavera intervention, they continued to use an experiential approach, when appropriate, in some teaching sessions.

A systematic review of universal school-based prevention programmes for alcohol misuse in young people included 53 trials, most of which were conducted in North America [36]. Only six trials were conducted in Europe. A majority of studies evaluated generic interventions rather than alcohol-specific interventions. From the 11 studies that evaluated alcohol-specific interventions, only six studies found significant beneficial effects of the interventions. Similarly, 14 of the 39 trials that evaluated generic programmes reported significantly beneficial effects of the programmes with regards to some of the measures of alcohol use. The generic prevention programmes based on psychosocial or developmental approaches were more likely to report positive results, although with effect sizes that were usually modest.

A systematic review of school-based programmes for preventing smoking identified 49 randomized controlled trials [9]. Globally, a significant effect of the interventions in preventing young people from starting smoking was found at longer than one year. Programmes that used a social competence approach and those that combined a social competence with a social influence approach were found to be more effective than other programmes. No overall effect was found, however, at one year or less, except for programmes that taught young people to be socially competent and to resist social influences.

Some limitations to the present study should, however, be pointed out. The multiplicity of outcomes could introduce a multiple comparison bias. To compensate for this, all the outcomes were introduced in adjustment model A2. Secondly, after randomization, the two groups did not follow programmes of the same intensity. The control prevention programme had a more limited duration than the Primavera programme and was not prolonged beyond the first year. Thirdly, the intended number of students for the two groups could not be reached, and attrition was higher than expected after the students’ transition from primary to secondary school. The transition from primary to secondary school, which proved to be the most delicate stage in the follow-up of the samples, could partly explain this high attrition. The fact that some of the students went to non-state secondary schools or moved away is the most plausible hypothesis identified by the school nurses for the reduction in numbers observed. Fourthly, the last assessment (time point 3) could not be conducted at the end of the third year of the programme due to the health crisis brought on by the SAR-COV2 pandemic. Finally, regulatory constraints did not enable a follow-up of the students’ individual evolution between assessment time points; as in most studies of this type [8,19,35,37], a cluster randomization was the only choice, and this reduced the power of the statistical analyses.

## 5. Conclusions

This randomized controlled cluster study enabled the efficacy of the Primavera programme in reducing alcohol use among schoolchildren to be demonstrated: after adjustment, children from the Primavera group were less likely to report current alcohol use at the end of the first year and past-month alcohol use at the end of the second year compared to those from the control group The prevention programme was initiated at the end of the final year of primary school with 10-year-old children and continued for another 2 years. For the first time in France, using a rigorous methodology, it has been possible to demonstrate the usefulness of implementing a multimodal, generic, universal, pluri-annual, prevention programme based on an experiential approach.

## Figures and Tables

**Figure 1 ijerph-18-03852-f001:**
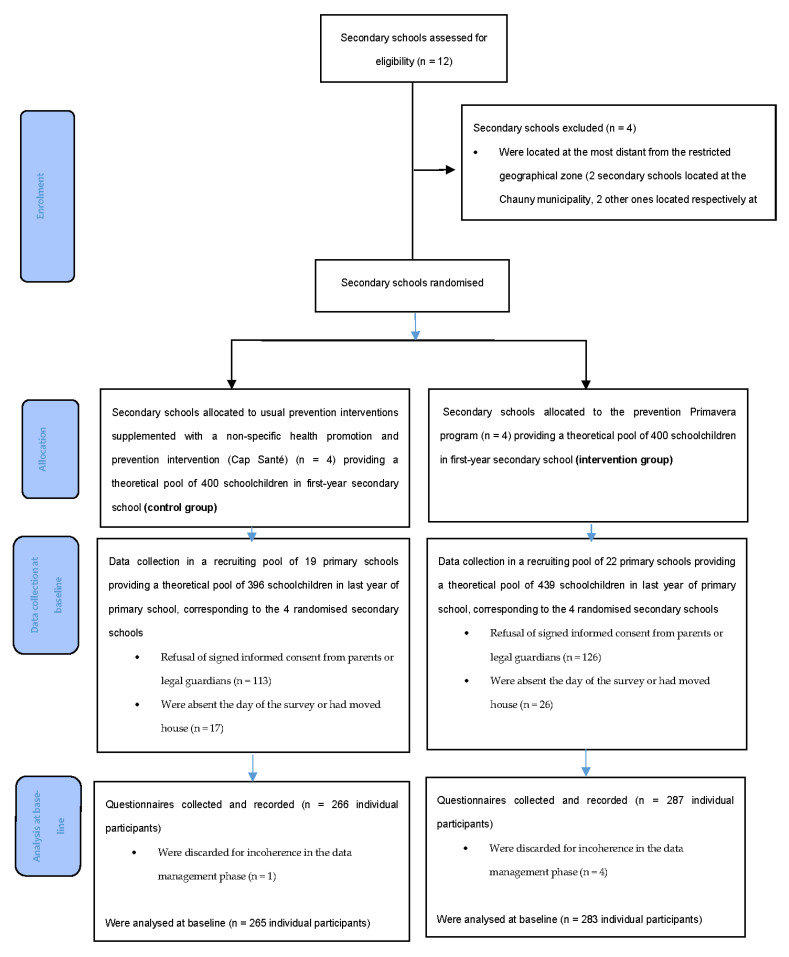
Flow diagram.

**Table 1 ijerph-18-03852-t001:** Results of the bivariate and multivariate logistic regression analysis at baseline and across follow-up time points 1 and 2.

Primary and Secondary Outcomes	Intervention Group Baseline (*n* = 283) Follow-Up Time Point 1 (*n* = 273) Follow-Up Time Point 2 (*n* = 197) *n* (%)	Control Group Baseline (*n* = 265) Follow-Up Time Point 1 (*n* = 255) Follow-Up Time Point 2 (*n* = 163) *n* (%)	OR (95% CI)	ORa1 (95% CI)	ORa2 (95% CI)
**Consumption**					
***Tobacco lifetime use***					
*Baseline*	17(6)	18(6.8)	0.88 [0.44–1.75]	0.56 [0.20–1.43]	0.38 [0.09–1.43]
*Follow-up time point 1*	16(6)	19(7.5)	0.77 [0.38–1.54]	0.51 [0.18–1.39]	0.35 [0.07–1.64]
*Follow-up time point 2*	21(10.7)	16(9.8)	1.10 [0.55–2.21]	0.79 [0.26–2.28]	1.12 [0.23–5.25]
***Last-month alcohol use***					
*Baseline*	20(7.1)	14(5.4)	1.36 [0.67–2.80]	1.23 [0.54–2.87]	1.22 [0.44–3.49]
*Follow-up time point 1*	16(5.9)	12(4.7)	1.26 [0.59–2.79]	1.07 [0.37–3.15]	1.95 [0.45–9.23]
*Follow-up time point 2*	13(6.7)	13(8)	0.83 [0.37–1.85]	**0.10 [0.02–0.42]**	**0.07 [0.01–0.66]**
***Last-month tobacco use***					
*Baseline*	5(1.8)	7(2.7)	0.66 [0.19–2.09]	NA	NA
*Follow-up time point 1*	5(1.8)	8(3.1)	0.58 [0.17–1.75]	NA	NA
*Follow-up time point 2*	6(3.1)	5(3.1)	1.00 [0.30–3.54]	NA	NA
***Current alcohol use***					
*Baseline*	71(25.3)	54(21.3)	1.24 [0.83–1.87]	1.04 [0.63–1.70]	1.14 [0.63–2.07]
*Follow-up time point 1*	60(22.3)	65(25.9)	0.82 [0.55–1.23]	**0.50 [0.27–0.88]**	**0.39 [0.18–0.78]**
*Follow-up time point 2*	52(26.9)	55(34)	0.72 [0.45–1.13]	0.63 [0.32–1.20]	0.68 [0.32–1.44]
**◆ Psychosocial skills**					
***PSS1***					
*Baseline*	118(42.9)	112(43.3)	0.98 [0.70–1.38]	0.90 [0.59–1.38]	0.96 [0.58–1.57]
*Follow-up time point 1*	142(53.4)	144(57.1)	0.86 [0.61–1.21]	0.70 [0.42–1.15]	0.72 [0.41–1.27]
*Follow-up time point 2*	120(62.2)	105(66)	0.85 [0.54–1.31]	0.81 [0.43–1.52]	0.88 [0.42–1.85]
***PSS2***					
*Baseline*	130(46.8)	152(58.9)	**0.61 [0.43–0.86]**	**0.60 [0.39–0.92]**	**0.59 [0.36–0.96]**
*Follow-up time point 1*	152(56.9)	147(58.8)	0.93 [0.65–1.31]	1.05 [0.64–1.75]	1.07 [0.61–1.88]
*Follow-up time point 2*	107(55.2)	74(46.5)	1.41 [0.93–2.16]	1.34 [0.73–2.51]	1.47 [0.72–3.09]
***PSS3***					
*Baseline*	170(61.4)	155(60.1)	1.06 [0.75–1.50]	1.07 [0.70–1.64]	1.21 [0.74–2.00]
*Follow-up time point 1*	178(66.9)	150(59.5)	1.38 [0.96–1.97]	1.23 [0.74–2.07]	1.23 [0.69–2.21]
***Follow-up time point 2***	115(59.9)	112(71.3)	**0.60 [0.38–0.94]**	**0.52 [0.27–0.99]**	0.55 [0.26–1.15]
***PSS4***					
*Baseline*	117(42.9)	146(57.3)	**0.56 [0.40–0.79]**	**0.54 [0.35–0.82]**	0.61 [0.37–1.01]
*Follow-up time point 1*	133(51)	136(54)	0.89 [0.63–1.25]	0.88 [0.53–1.46]	0.88 [0.48–1.61]
*Follow-up time point 2*	99(51.6)	75(47.5)	1.18 [0.77–1.80]	1.47 [0.79–2.79]	1.88 [0.88–4.15]
***PSS5***					
*Baseline*	136(49.3)	155(59.6)	0.66 [0.47–0.93]	0.62 [0.40–0.96]	0.67 [0.41–1.10]
*Follow-up time point 1*	145(55.1)	140(56.5)	0.95 [0.67–1.34]	1.05 [0.62–1.80]	1.14 [0.63–2.10]
*Follow-up time point 2*	106(55.5)	103(64.8)	0.68 [0.44–1.04]	0.63 [0.33–1.21]	0.64 [0.29–1.39]
***PSS6***					
*Baseline*	126(45.5)	156(60.2)	**0.55 [0.39–0.78]**	**0.56 [0.36–0.85]**	0.62 [0.38–1.01]
*Follow-up time point 1*	140(52.6)	137(54.2)	0.94 [0.67–1.33]	0.91 [0.55–1.50]	0.92 [0.53–1.60]
*Follow-up time point 2*	89(45.9)	83(52.2)	0.78 [0.51–1.18]	0.88 [0.47–1.63]	0.83 [0.40–1.70]
**⦿ *Capacity to say "no" to a smoke for refusing a friend***					
*Baseline*	227(82.5)	221(84)	0.90 [0.57–1.41]	0.84 [0.47–1.49]	2.36 [0.70–8.81]
*Follow-up time point 1*	229(84.8)	219(85.9)	0.92 [0.56–1.49]	0.66 [0.33–1.32]	0.41 [0.13–1.28]
*Follow-up time point 2*	169(87.6)	138(87.3)	1.02 [0.54–1.92]	0.69 [0.28–1.72]	3.65 [0.41–4.36]
**⦿ *Capacity to say "no" to a drink for refusing a friend***					
*Baseline*	203(73.6)	206(78.6)	0.76 [0.51–1.12]	0.72 [0.44–1.17]	0.49 [0.20–1.14]
*Follow-up time point 1*	218(81.3)	203(79.9)	1.10 [0.71–1.69]	1.00 [0.53–1.88]	1.78 [0.60–5.63]
*Follow-up time point 2*	157(80.9)	137(85.6)	0.71 [0.40–1.25]	0.63 [0.28–1.43]	0.70 [0.15–3.31]
**✪ Denormalization of tobacco or alcohol use**					
***Denormalization of tobacco use at home***					
*Baseline*	264(96.4)	239(94.5)	1.55 [0.68–3.65]	NA	NA
*Follow-up time point 1*	260(95.9)	246(96.9)	0.77 [0.29–1.93]	NA	NA
*Follow-up time point 2*	185(94.9)	157(97.5)	0.47 [0.13–1.44]	NA	NA
***Denormalization of alcohol use at home***					
*Baseline*	225(80.6)	214(86.6)	0.64 [0.40–1.02]	NA	NA
*Follow-up time point 1*	231(85.6)	221(87.4)	0.86 [0.52–1.42]	NA	NA
*Follow-up time point 2*	165(85.1)	143(88.8)	0.72 [0.38–1.33]	NA	NA
***Denormalization of tobacco use at school***					
*Baseline*	220(82.4)	196(80.3)	1.15 [0.73–1.79]	1.13 [0.67–1.90]	0.89 [0.43–1.83]
*Follow-up time point 1*	225(84)	208(82.2)	1.13 [0.72–1.79]	1.71 [0.92–3.23]	**4.11 [1.60–11.33]**
*Follow-up time point 2*	156(80)	123(76.4)	1.24 [0.74–2.05]	1.41 [0.68–3.03]	1.79 [0.54–6.24]
***Denormalization of alcohol use at school***					
*Baseline*	202(74.3)	166(69.7)	1.25 [0.85–1.85]	1.52 [0.95–2.46]	1.60 [0.81–3.22]
*Follow-up time point 1*	179(66.8)	187(73.9)	0.71 [0.48–1.04]	0.83 [0.49–1.40]	**0.44 [0.21–0.90]**
*Follow-up time point 2*	130(67)	104(64.6)	1.11 [0.72–1.73]	1.51 [0.77–3.01]	1.22 [0.45–3.51]
**❖ Peer support ****					
***PS1***					
*Baseline*	205(74)	200(76)	0.90 [0.61–1.32]	1.07 [0.66–1.74]	1.09 [0.63–1.90]
*Follow-up time point 1*	179(66.5)	185(72.8)	0.74 [0.51–1.08]	0.72 [0.42–1.24]	0.79 [0.41–1.53]
*Follow-up time point 2*	118(61.1)	102(63.8)	0.89 [0.58–1.38]	1.02 [0.54–1.99]	0.88 [0.38–2.10]
***PS2***					
*Baseline*	191(68.7)	189(71.9)	0.86 [0.59–1.24]	1.12 [0.71–1.77]	1.21 [0.71–2.09]
*Follow-up time point 1*	185(68.5)	180(70.9)	0.89 [0.62–1.30]	0.98 [0.58–1.68]	0.88 [0.44–1.78]
*Follow-up time point 2*	108(56)	90(56.6)	0.97 [0.64–1.49]	0.91 [0.49–1.73]	0.80 [0.35–1.86]
***PS3***					
*Baseline*	223(80.5)	212(80.9)	0.97 [0.63–1.49]	0.96 [0.55–1.67]	0.91 [0.48–1.75]
*Follow-up time point 1*	208(77.9)	194(76.1)	1.11 [0.74–1.67]	0.92 [0.51–1.64]	0.90 [0.44–1.84]
*Follow-up time point 2*	138(71.1)	121(75.6)	0.79 [0.49–1.28]	0.78 [0.39–1.59]	0.84 [0.34–2.08]

◆ “Most often”, “often” compared to “never”, “almost never” or “sometimes” (recoded 1 vs. 0). ⦿ “Very easy” or “rather easy to refuse” compared to “I do not know”, “rather difficult or very difficult to refuse” (recoded 1 vs. 0). ✪ “Disagree completely”, “rather disagree” compared to “rather agree”, “completely agree” or “neither agree nor disagree” (recoded 1 vs. 0). ❖ “Completely agree” or “rather agree” compared to “neither agree nor disagree”, “rather disagree”, “disagree completely” (recoded 1 vs. 0). NA = Not applicable. The conditions for the validity of multivariate logistic analyses were not met. The predefined adjustment variables were the following: gender, age, three social-economic variables (individual bedroom, two cars or more in the household, trip abroad during the last holiday), and a weighting variable for each school to take the magnitude of the numbers analysed for each school into account. The other potential confounding variables for baseline were the following: outings to parks and museums, no more than 2 hours spent watching screens during the week, good school results and parental alcohol consumption. The other potential confounding variables for follow-up time point 1 were the following: outings to parks and museums, under two hours watching screens in the week, good school results, talking about alcohol at home with parents, and remembering having received information in school about alcohol, tobacco or another substance. The other potential confounding variables for follow-up time point 2 were the following: having a parent using tobacco or alcohol and remembering having received information in school about alcohol, tobacco or another substance. Multivariate logistic regression model A1: adjusting on predefined adjustment variables and on other variables with an OR significantly different from 1 in the bivariate logistic regression for the follow-up time point considered. Multivariate logistic regression model A2: identical to model A1 but with the introduction of all the other primary and secondary outcomes.

## Data Availability

The data that support the findings of this study are available from the corresponding author, H.-J.A., upon reasonable request.

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
