# Peer review of "The Efficacy of Primavera, a Prevention Programme on Alcohol and Tobacco Use among 10–12-Year-Old Schoolchildren: A Randomized Controlled Cluster Study"

_ijerph, 2021, doi:10.3390/ijerph18083852_

Round 1
Reviewer 1 Report
Revision for the submission no: ijerph-1102269
This is a well-written manuscript presenting findings from a randomized controlled cluster study aimed at assessing the effects of a new prevention program (Primavera) compared to a control subject to standard intervention among French schoolchildren from 10 to 12 years. As alcohol and tobacco use remains a major health problem in Europe efficient prevention strategies are of utmost need. The presented study is based on a substantial dataset (over 500 participants) and provides a good quality statistical analysis, as well as a description of the results. The biggest advantage of this paper is the fact, that the findings from this study may have interesting practical implications.
However, there are two major shortcomings – insufficient information regarding the current situation in France regarding alcohol and tobacco consumption in children, and a lack of proper discussion of the study’s results. The latter is very disappointing, as the study manifests meaningful potential to shape future preventive actions.
Authors are asked to broaden the discussion section. Special attention should be paid to compare findings from the presented study with results of other similar interventions.
Author Response
Thank you for these relevant suggestions.
We added information regarding the current situation in France in the first paragraph of the introduction: “According to the Health Behaviour in School-aged Children (HBSC) and the European School Project on Alcohol and other Drugs (ESPAD) surveys carried out simultaneously for the first time in France in 2018 (EnCLASS survey) among 20,000 middle and high school students aged 11 to 18, alcohol is the first psychoactive substance children experiment with [5]. The number of students found to have drunk at least one alcoholic beverage in their life in Grade 6 (44.3%) clearly indicates that this initiation to alcohol begins in primary school. However, less than one in ten middle school students in Grade 6 (8.6%) reported that they have drunk in the past month. The number of students using tobacco is significantly lower at the beginning of the middle school (1.4% last month smoking), although 7.6% reported that they had already smoked. As with lifetime use of alcohol, the «middle school years» seem to mainly be a phase where young adolescents are introduced to the tobacco. For the two substances studied, boys experiment more than girls. Alcohol is still the first substance used during adolescence, followed by tobacco. Moving from Grade 9 (last grade of middle school) to Grade 10 (first grade of high school) is characterised by pursuing experimenting. As is true for middle school, alcohol is the most commonly used substance among high school students (85.0%), considerably more than tobacco (53.0%).”
We also introduced a global description of the Primavera prevention program as an appendix in the supplementary materials, and broadened the discussion section. In particular, we added:
“In addition, the program is implemented across the primary-secondary school transition, a critical period in terms of emotional, social, and academic adjustment [12-14]. This program is in current use in French schools by several trained teams. An informal qualitative exploration accompanying the study showed it has been well received and accepted by educational teams, children and parents. For example, teachers reported that, following their experience with the Primavera intervention, they continued to use an experiential approach when appropriate in some teaching sessions. Others suggested to extend the prevention program for internet and gaming addiction.
A systematic review of universal school-based prevention programs for alcohol misuse in young people included 53 trials, most of which were conducted in North America [36]. Only six trials were conducted in Europe. A majority of studies evaluated generic interventions rather than alcohol-specific interventions. From the 11 studies that evaluated alcohol-specific interventions, only 6 studies six studies found significant beneficial effects of the interventions. Similarly, 14 of the 39 trials that evaluated generic programmes reported significantly beneficial effects of the programmes with regards to some of the measures of alcohol use. The generic prevention programmes based on psychosocial or developmental approaches were more likely to report positive results, although with effect sizes that were usually modest.
A systematic review of school-based programmes for preventing smoking identified 49 randomised controlled trials [7]. Globally, a significant effect of the interventions in preventing young people from starting smoking was found at longer than one year. Programmes that used a social competence approach and those that combined a social competence with a social influence approach were found to be more effective than other programmes. No overall effect was found however at one year or less, except for programmes which taught young people to be socially competent and to resist social influences.”
Reviewer 2 Report
I was very interested to have the opportunity to consider a paper that would consider the efficacy of a schools-based prevention programme. Such an evaluation would have considerable 'real world' interest, with potential to impact on practice.
Unfortunately, the paper as it stands provides insufficient information about the programme and its contexts to be able to understand what has been done, with what purpose, in what specific contexts and with whom. A primary prevention programme would take a different approach from a secondary programme. What needs assessment was undertaken? How far were parents involved? What effects did socio-environmental contexts have? How far were parents involved in schools-based programmes already?
Some of the introduction is unhelpfully vague - what are you suggesting are the 'real motives' to use?
Also, there's lots of evidence to suggest that transition points, notably from primary to secondary school provide opportunities for impact that might not exist at other points. Did consideration of that form part of the intervention strategy?
More information about all of the above would massively improve this paper.
I hope that's helpful.
Author Response
I was very interested to have the opportunity to consider a paper that would consider the efficacy of a schools-based prevention programme. Such an evaluation would have considerable 'real world' interest, with potential to impact on practice.
Unfortunately, the paper as it stands provides insufficient information about the programme and its contexts to be able to understand what has been done, with what purpose, in what specific contexts and with whom. A primary prevention programme would take a different approach from a secondary programme. What needs assessment was undertaken? How far were parents involved? What effects did socio-environmental contexts have? How far were parents involved in schools-based programmes already?
R: Thank you very much for noting the lack of sufficient information about the program. We have introduced a global description of the Primavera prevention program as an appendix in the supplementary materials. We also added this in the discussion section: “An informal qualitative exploration accompanying the study showed it has been well received and accepted by educational teams, children and parents. For example, teachers reported that, following their experience with the Primavera intervention, they continued to use an experiential approach when appropriate in some teaching sessions. Others suggested to extend the prevention program for internet and gaming addiction.”
Some of the introduction is unhelpfully vague - what are you suggesting are the 'real motives' to use?
R: We have significantly changed the introduction section, in particular we wave deleted the problematic sentence: “Alcohol and tobacco use behaviours are related to individual factors, such as poor awareness of the specific risks involved in the medium or long term, or the real motives leading young people to indulge in them.”
Also, there's lots of evidence to suggest that transition points, notably from primary to secondary school provide opportunities for impact that might not exist at other points. Did consideration of that form part of the intervention strategy?
R: Actually, yes it was. It was not mentioned in the first version of the manuscript, as we didn’t find published evidence to back this strategy. Thanks to your comment, (in particular the use of the term transition, we were able to find proper references (Engels et al. Students' academic and emotional adjustment during the transition from primary to secondary school: A cross-lagged study. Journal of School Psychology 2019, 76, 140-158; Uka, A. The Effect of Students’ Experience with the Transition from Primary to Secondary School on Self-Regulated Learning and Motivation. Sustainability 2020, 12, 8519; White, J. Evidence Review: supporting children's mental health and wellbeing at transition from primary to secondary school; 2020). We therefore made the point in the introduction (“Continuity in action over several school years, especially during transition between primary and secondary school, could also be a factor predictive of efficacy [7,10-14].”), as in the discussion section (“In addition, the program is implemented across the primary-secondary school transition, a critical period in terms of emotional, social, and academic adjustment [12-14].”)
More information about all of the above would massively improve this paper.
I hope that's helpful.
R: Yes, we think your suggestions allowed the manuscript to greatly improve. Thank you very much.
Round 2
Reviewer 2 Report
I'm really sorry but I still think this paper needs more work. I'm still struggling to assess the paper - though I think it could be useful - as there is still too little about the programme here. There is more but confusingly in the Discussion section rather than upfront in the Introduction.
What does the author mean that there is a global description in supplementary materials? I don't, I think, have access to those and the description needs to be in the paper so the paper can have a coherent argument.
The new section in the introduction from line 39 just seems to have been dropped in - there needs to be some joining up - 'in response to this, the Primavera programme was developed which...'
Why is gaming and addiction introduced in line 373. How is that recommendation from an informal qualitative exploration' (what's that?) relevant to the focus of this paper, which is to consider the effectiveness of this programme?
From what's presented, I'm not able to see how the very general statement that this programme 'is effective on alcohol use among preadolescents' can be validated. In any case, what does 'effective on alcohol use' mean? Also, what's a 'preadolescent' for the purposes of this study? Are these definitely the target group?
I would suggest stepping back and really looking at what this paper is about. I am sure that there has been some good work done here, with potentially useful learning.
Author Response
I'm really sorry but I still think this paper needs more work. I'm still struggling to assess the paper - though I think it could be useful - as there is still too little about the programme here. There is more but confusingly in the Discussion section rather than upfront in the Introduction.
R: You may not have been aware of the remarks made by the other reviewer, which could explain your confusion.
What does the author mean that there is a global description in supplementary materials? I don't, I think, have access to those and the description needs to be in the paper so the paper can have a coherent argument.
R: We understand the misunderstanding if you had no access to the supplementary materials. Please note it was however uploaded as a separate file along with the manuscript in the submission process. In any case, following your advice, we moved the 670-word long description of the Primavera progamme in the 2.4 section of the manuscript. We agree the reader will have an easier access to this description directly in the manuscript rather than in a separate file. We hope the editorial office will accept the ensuing increase in length of the manuscript.
The new section in the introduction from line 39 just seems to have been dropped in - there needs to be some joining up - 'in response to this, the Primavera programme was developed which...'
R: Thank you for noticing this. We corrected and completed the introduction in several ways, including the suggested transition sentence in the beginning of the last paragraph of the introduction.
Why is gaming and addiction introduced in line 373. How is that recommendation from an informal qualitative exploration' (what's that?) relevant to the focus of this paper, which is to consider the effectiveness of this programme?
R: Thank you for noticing this. We agree the reference to internet and gaming addiction was irrelevant in this context. We deleted it.
From what's presented, I'm not able to see how the very general statement that this programme 'is effective on alcohol use among preadolescents' can be validated. In any case, what does 'effective on alcohol use' mean? Also, what's a 'preadolescent' for the purposes of this study? Are these definitely the target group?
R: We deleted all reference to preadolescence, and used instead the term “schoolchildren”. We tried to be more specific when stating the effectiveness of the programme.
In the abstract: “. After adjustment, children from the Primavera group were less likely to report current alcohol use at the end of the first year and past-month alcohol use at the end of the second year compared to those from the control group […]The Primavera prevention program is proven shown to be effective on in reducing alcohol use among schoolchildren.
And in the conclusion section: “This randomised controlled cluster study enabled the efficacy of the Primavera program in reducing alcohol use among schoolchildren to be demonstrated: after adjustment, children from the Primavera group were less likely to report current alcohol use at the end of the first year and past-month alcohol use at the end of the second year compared to those from the control group.”
I would suggest stepping back and really looking at what this paper is about. I am sure that there has been some good work done here, with potentially useful learning.
R: Thank you very much for this in-depth assessment of our manuscript. We hope that you will find this second revision appropriately improved and satisfactory.
